# Essential childhood immunization in 43 low- and middle-income countries: Analysis of spatial trends and socioeconomic inequalities in vaccine coverage

**Anna Dimitrova**[1] *, **Gabriel Carrasco-Escobar**[1,2], **Robin Richardson**[3,4], **Tarik Benmarhnia**[1]

**1** Scripps Institution of Oceanography, University of California, San Diego, California, United States of America, **2** Health Innovation Laboratory, Institute of Tropical Medicine "Alexander von Humboldt", Universidad Peruana Cayetano Heredia, Lima, Peru, **3** Mailman School of Public Health, Columbia University, New York, New York, United States of America, **4** Rollins School of Public Health, Emory University, Atlanta, Georgia, United States of America

* adimitrova@ucsd.edu

**Data Availability Statement:** The data underlying the results presented in the study are available from the Demographic and Health Surveys (DHS) website (https://dhsprogram.com/data/). The authors do not have the right to share DHS data.

## Abstract

### Background

Globally, access to life-saving vaccines has improved considerably in the past 5 decades. However, progress has started to slow down and even reverse in recent years. Understanding subnational heterogeneities in essential child immunization will be critical for closing the global vaccination gap.

### Methods and findings

We use vaccination information for over 220,000 children across 1,366 administrative regions in 43 low- and middle-income countries (LMICs) from the most recent Demographic and Health Surveys. We estimate essential immunization coverage at the national and sub-national levels and quantify socioeconomic inequalities in such coverage using adjusted concentration indices. Within- and between-country variations are summarized via the Theil index. We use local indicator of spatial association (LISA) statistics to identify clusters of administrative regions with high or low values. Finally, we estimate the number of missed vaccinations among children aged 15 to 35 months across all 43 countries and the types of vaccines most often missed. We show that national-level vaccination rates can conceal wide subnational heterogeneities. Large gaps in child immunization are found across West and Central Africa and in South Asia, particularly in regions of Angola, Chad, Nigeria, Guinea, and Afghanistan, where less than 10% of children are fully immunized. Furthermore, children living in these countries consistently lack all 4 basic vaccines included in the WHO's recommended schedule for young children. Across most countries, children from poorer households are less likely to be fully immunized. The main limitations include subnational estimates based on large administrative divisions for some countries and different periods of survey data collection.

**Funding:** The author(s) received no specific funding for this work.

**Competing interests:** The authors have declared that no competing interests exist.

**Abbreviations:** BCG, bacille Calmette–Guerin; COVID-19, Coronavirus Disease 2019; DTP, diphtheria–tetanus–pertussis; EPI, Expanded Program on Immunization; FIC, full immunization coverage; HepB, hepatitis B vaccine; LISA, local indicator of spatial association; LMIC, low- and middle-income country; MCV, measles-containing vaccine; OPV, oral polio vaccine; PCV, pneumococcal conjugate vaccine; PSU, primary sampling unit; USAID, United States Agency for International Development.

## Conclusions

The identified heterogeneities in essential childhood immunization, especially given that some regions consistently are underserved for all basic vaccines, can be used to inform the design and implementation of localized intervention programs aimed at eliminating child suffering and deaths from existing and novel vaccine-preventable diseases.

## Author summary

### Why was this study done?

- Despite global efforts to improve child immunization rates in low- and middle-income countries (LMICs), progress has slowed down and even reversed in recent years.

- Identifying hard-to-reach populations will be critical for closing the vaccination gap.

- Socioeconomic disparities in child immunization coverage have been mostly studied at the national level.

### What did the researchers do and find?

- We analyzed survey data from 43 LMICs and investigated disparities in child vaccination coverage at the subnational level and across socioeconomic groups.

- We identified geographical regions in Africa and Asia where levels of childhood vaccination are particularly low.

- Across most countries, children from poorer households are less likely to be fully immunized and a large number of children miss all 4 essential vaccines recommended by the World Health Organization.

### What do these findings mean?

- Large gaps in child immunization are found across and within countries, and among socioeconomic groups.

- More efforts are needed to ensure equitable access to essential vaccines in LMICs, where infectious diseases are among the leading causes of child death.

## Introduction

Vaccination is one of the most cost-effective interventions in public health [1,2] that has led to the control and eradication of certain highly lethal infectious diseases [3–5]. Despite notable efforts to improve access to essential vaccines globally [6,7], the benefits have not been

distributed equally both across and within countries [7,8]. Child immunization against common infectious diseases such as measles, polio, and diphtheria, has become routine practice in high-income countries where millions of lives have been saved as a result [4,9]. In contrast, the burden of such diseases remains far too high in low- and middle-income countries (LMICs). Globally, 1.5 million child deaths under the age of 5 are attributed to vaccine-preventable diseases every year and the vast majority of these occur in sub-Saharan Africa and South Asia [10].

In 2021, the Immunization Agenda 2030 was launched with the aim of improving access to vaccines globally and ensuring higher vaccine equity [11]. As core targets, the agenda aspires to achieve at least 90% coverage of essential childhood vaccines in every country and reduce by 50% the number of entirely unvaccinated children. Meeting these ambitious goals will require a good understanding of which populations have been left behind and their barriers to receiving life-saving immunization. However, statistics about vaccine coverage are usually reported at the national level [6,7,12], which is likely to mask large inequalities both at the subnational level [13] and across socioeconomic groups [8]. Identifying regions with high shares of under- or unvaccinated children, where the risk of disease outbreaks is high, will be critical for closing the vaccination gap between poor and rich nations.

Previous studies that have investigated socioeconomic disparities in full immunization coverage (FIC) in LMICs have mainly focused on national-level disparities [14–16]. Several studies have explored subnational heterogeneities in child immunization coverage but these have been assessed for individual countries [17,18] or specific vaccines [13,19]. Still little is known about the presence of socioeconomic disparities in child immunization at the subnational level [20]. Moreover, the variety of definitions of FIC used in the literature and the range of age groups for which these have been assessed [20] make estimates reposted in previous studies incomparable. We add to the literature by presenting harmonized and spatially disaggregated estimates of full immunization coverage and wealth-related inequalities in such coverage across multiple LMICs.

We use detailed immunization data for over 220,000 children from 1,366 administrative regions in 43 countries. We estimate at the national and subnational levels the share of children who have received full immunization following the schedule for young children recommended in the WHO's Expanded Program on Immunization (EPI) [21]. We also quantify wealth-related inequalities in FIC at the national and subnational levels. A range of mapping techniques is used to reveal distinct spatial patterns in child immunization. Clusters of administrative regions characterized by low vaccine coverage, or a high degree of socioeconomic inequality are identified using a spatial association technique. Moreover, the exact type and number of essential vaccines that are missed per country are discerned from the data.

## Data and methods

This study is reported as per the Strengthening the Reporting of Observational Studies in Epidemiology (STROBE) guideline (S1 Table).

## Data source

This study uses data from the most recent round of the DHS, collected between 2014 and 2021 (S2 Table [22]). The DHS surveys are repeated cross-sectional surveys conducted in over 90 low- to middle-income countries and are a principal source of information on fertility, family planning, maternal and child health, and the provision of health services. The surveys also contain information about the socioeconomic profile of households, including the education level and occupation of household members and household wealth status. The surveys are funded

by the United States Agency for International Development (USAID) and were first launched in 1984 to improve the understanding of health and population trends in LMICs. The survey design is based on a two-step sampling procedure [23]. In the first stage, probability samples are drawn from an existing sample frame, usually the most recent census. The sampling frame is divided into subgroups (strata), which are typically geographic regions and urban/rural areas within each region. Within each stratum, primary sampling units (PSUs) are selected with probability proportional to the size within each stratum. In the second stage, a complete household listing is conducted in each PSU and a fixed number of households is selected via systematic sampling. Sampling weights are provided to adjust for differences in the probability of selection and interview. S1 Fig shows the location of PSUs in each country included in the analysis. The sampling procedure ensures that the samples are representative both at national and subnational levels (usually administrative level 1 or 2, e.g., region or province) and by urban/rural area. A detailed description of data collection and validation procedures are described elsewhere [23,24]. Key strengths of the DHS surveys are the high response rates (usually over 90%), their national coverage, the high quality of interviewer training, and the standardized data collection procedures, which allow comparisons across countries and over time [25]. The DHS surveys have become a valuable source of information in epidemiological research, with a wide range of applications for monitoring of prevalence, trends, and inequalities.

Data on essential vaccines are regularly collected as part of the DHS program. We focused on the vaccines initially recommended by WHO as part of the basic immunization schedule for young children, also known as the EPI [26]. These include 1 dose of bacille Calmette–Guerin (BCG) vaccine, 3 doses of diphtheria–tetanus–pertussis (DTP) vaccine, 3 doses of oral polio vaccine (OPV), and 1 measles-containing vaccine (MCV). Since the program was introduced in 1974 [27], additional vaccines have been added to the list, including hepatitis B vaccine (HepB), *Haemophilus influenzae* type b vaccine (Hib), rubella vaccine, pneumococcal conjugate vaccine (PCV), and rotavirus vaccine; however, these have been adopted by countries in their publicly funded immunization programs at a different pace. For comparability purposes, we focus on the 8 vaccine doses included in the initial EPI schedule.

The DHS recode files for children (KR) were retrieved for 43 countries for which detailed vaccine information was collected. All children under 5 years of age living with their biological mother were eligible to participate in the DHS survey. The vaccine information was collected only for children under 3 years of age who were alive at the time of the interview; therefore, we limited our analysis to this subset of children. Health cards were used to determine the immunization status of children. If the health card was missing or the information on the card was incomplete, mother's recall was used instead.

## Measures

We determined the immunization status of children based on the 4 vaccines included in the EPI schedule. Typically, BCG is administered shortly after birth, DTP 1–3 and OPV 1–3 are administered at 6, 10, and 14 weeks after birth, and MCV is administered between 9 and 13 months of age, depending on the national immunization schedule [28]. For comparability purposes and to allow for some catch-up vaccination, we restricted the sample to children 15 to 35 months of age. To test the robustness of our findings, we also assessed vaccine coverage for children 24 to 35 months of age.

Complete vaccination information, via vaccination cards or mothers' recall, was available for 98% of children. The pooled sample consisted of 221,693 children from 1,366 administrative divisions. The administrative divisions can be provinces, districts, or other divisions,

depending on the administrative level at which DHS samples are representative. We generated a series of binary variables indicating whether the child had received each of the vaccines at any age. We also generated a binary variable for full immunization status—children reporting no vaccinations or any vaccine combination other than the full course were categorized as not fully immunized.

We additionally retrieved information about the household's socioeconomic situation, which was used to examine the presence of wealth-related inequalities in basic childhood immunization. We used the wealth index available in DHS surveys to determine the socioeconomic situation of households. In DHS, the wealth index is constructed based on principal component analysis and combining information about the household's ownership of selected assets, building characteristics, overcrowding, and the presence of domestic servants [29]. Different items may be included depending on data availability for each country and survey round. The wealth index indicates a household's relative socioeconomic position in relation to other households in the same country. Household wealth is generally preferred to other indicators of economic status such as income or consumption expenditure that are often unavailable or unreliable in the context of LMICs [30].

Socioeconomic inequalities in the full immunization status of children were quantified using the concentration index [31,32], which has been previously used in the literature to measure the magnitude of income-related inequalities in various health indicators [33–35], including childhood immunization [15,16]. The concentration index is measured in relation to the concentration curve, which plots on the $x$-axis the cumulative share of the children ranked by socioeconomic position (from the lowest to the highest), and on the $y$-axis the cumulative child vaccination coverage (S2 Fig). If vaccination coverage is equally distributed among all children ranked by socioeconomic position, the concentration curve will coincide with the 45˚ line. The concentration index measures twice the area between the 45˚ line and the concentration curve. The range of the concentration index is from −1 to +1, with negative values signifying the concentration of the relevant health variable among the lower socioeconomic groups and positive values indicating the opposite. The larger the absolute value of the index is, the greater the degree of inequality. A value of zero indicates no socioeconomic differential. With binary health variables, which is the case in this study, the use of the CI to measure health inequalities could be problematic [36]. In particular, the CI values are bounded between μ-1 and 1-μ [37], where μ is the mean of the health variable among the population, which makes the comparison of populations with different mean health levels problematic [37,38]. Moreover, the CI may result in different rankings depending on whether it is estimated with respect to health or ill health [39]. The choice of measurement scale for the health variable also affects the measured degree of inequality [38].

Different correction procedures have been proposed to deal with the above issues [36]. The most common when analyzing binary health variables are Wagstaff's [40] and Erreygers' [38] correction procedures. For our analysis, we use both Wagstaff's (W) and Erreygers' (E) adjusted concentration indices to measure socioeconomic inequalities in childhood immunization. The "conindex" package [41] in Stata 16 was used to estimate W and E. Sampling weights were applied in the calculation of both indices. More information about the adjustment procedures is provided elsewhere [37,38].

We additionally use the Theil index [42] as a summary measure for the amplitude of disparity in vaccination rates, W and E, within and between countries. For this purpose, we use the "ineqdeco" package [43] in Stata 16. The subnational level immunization rates are used as input data and grouped by country to decompose the Theil index into within-country and between-country inequality. A Theil index of 0 indicates perfect equality, whereas higher Theil index values indicate a higher degree of inequality. The Theil index is widely used as a

summary measure for the amplitude of within-group and between-group inequalities, including for health outcomes [44–46].

## Mapping subnational heterogeneities

We generated national and subnational estimates of FIC from the individual data. W and E values were also estimated at the national and subnational levels. Up-to-date subnational boundaries were retrieved from the DHS spatial data repository (https://spatialdata.dhsprogram.com/home/). Sampling weights were included in all aggregation procedures. We mapped the subnational heterogeneities in vaccine coverage and the corresponding W and E values for all eligible countries.

Further analysis was conducted to determine the type of vaccinations that are most often missed by children with incomplete immunization status. The intersecting sets of missed vaccines were examined using the "UpSetR" package [47] in R software v.4.0.1 [48]. The vaccine (or vaccine combinations) most often missed per country were discerned from the UpSet plot. The number of missed vaccines per country was then calculated by applying UN annual population estimates from 2020. The estimated number of missed vaccinations refers to children between 15 and 35 months of age. It should be noted that the reported numbers of missed vaccinations represent a rough estimate since we assume that vaccine coverage has remained unchanged from the time of the survey data collection, which ranges by country from 2014 to 2021.

## Spatial autocorrelation analysis

We then conducted spatial autocorrelation analysis to identify administrative regions where values are strongly associated with one another. We used a local *Getis-Ord Gi** statistic, a type of local indicator of spatial association (LISA), with a first-order queen contiguity-based weighted neighborhood structure. Under the contiguity criterion, 2 administrative regions are first-order neighbors if they share a common border. The tool is used to identify spatial clusters of high and low values and the corresponding level of statistical significance. Additionally, adjustment for false discovery rate is made to prevent bias due to multiple and dependent tests [49]. Spatial associations among administrative regions were visualized via maps. The spatial data processing and visualizations were performed in software v.4.0.1 [48]. Furthermore, codes to reproduce all results are available at the following link: https://github.com/benmarhnia-lab/vaccines_ineq.

## Results

Overall, large disparities in FIC can be seen across countries (Table 1). Rwanda has the highest immunization rate among the countries included in the analysis (95%). Albania and Bangladesh have also reached 90% immunization rates. In two-thirds of the countries included in the analysis, immunization rates of 50% and higher have been achieved but in 4 countries (Guinea, Chad, Angola, and Nigeria) less than a third of children are fully immunized.

At the subnational level, wide heterogeneities are observed as well (Fig 1 and S5 Table). Some of the lowest vaccination rates are observed in Africa, particularly in parts of Angola, Chad, Nigeria, Guinea, and Mali, where less than 10% of children are fully vaccinated. Similarly, very low levels of immunization are observed in southwestern Afghanistan and northeastern India.

The Theil index shows that the disparities in childhood immunization are greater regarding FIC than individual vaccines (Table 2). The decomposed Theil index further shows that the within-country and between-country disparities are equally pronounced. Moreover, the

**Table 1. National estimates of FIC, Wagstaff's index of inequality (W), and Erreygers' index of inequality (E) for 43 countries.**

| Country | Sample size | FIC | Rank FIC | W | Lower bound | Upper bound | Rank W | E | Lower bound | Upper bound | Rank E |
|---|---|---|---|---|---|---|---|---|---|---|---|
| Afghanistan | 10,597 | 0.421 | 10 | 0.133 | 0.111 | 0.155 | 17 | 0.130 | 0.108 | 0.152 | 15 |
| Albania | 865 | 0.906 | 41 | −0.121 | −0.253 | 0.010 | 20 | −0.041 | −0.086 | 0.003 | 34 |
| Angola | 4,797 | 0.282 | 3 | 0.410 | 0.376 | 0.445 | 1 | 0.333 | 0.305 | 0.361 | 1 |
| Armenia | 595 | 0.889 | 40 | −0.162 | −0.310 | −0.015 | 16 | −0.064 | −0.122 | −0.006 | 25 |
| Bangladesh | 2,883 | 0.907 | 42 | 0.131 | 0.059 | 0.204 | 18 | 0.044 | 0.020 | 0.069 | 33 |
| Benin | 4,283 | 0.553 | 17 | 0.191 | 0.157 | 0.226 | 12 | 0.189 | 0.155 | 0.223 | 10 |
| Burundi | 4,319 | 0.841 | 39 | −0.016 | −0.064 | 0.031 | 43 | −0.009 | −0.034 | 0.016 | 43 |
| Cambodia | 2,487 | 0.802 | 34 | 0.329 | 0.273 | 0.385 | 4 | 0.209 | 0.173 | 0.244 | 9 |
| Cameroon | 3,090 | 0.520 | 14 | 0.261 | 0.221 | 0.300 | 7 | 0.260 | 0.220 | 0.300 | 6 |
| Chad | 5,172 | 0.260 | 2 | 0.086 | 0.050 | 0.122 | 26 | 0.066 | 0.038 | 0.094 | 22 |
| Egypt | 5,569 | 0.415 | 9 | 0.088 | 0.058 | 0.119 | 25 | 0.086 | 0.056 | 0.116 | 19 |
| Ethiopia | 1,801 | 0.400 | 7 | 0.298 | 0.245 | 0.350 | 5 | 0.286 | 0.235 | 0.336 | 4 |
| Ghana | 1,987 | 0.753 | 30 | −0.017 | −0.076 | 0.042 | 42 | −0.013 | −0.056 | 0.031 | 42 |
| Guatemala | 4,221 | 0.823 | 37 | 0.089 | 0.044 | 0.135 | 24 | 0.052 | 0.025 | 0.079 | 28 |
| Guinea | 2,135 | 0.247 | 1 | 0.195 | 0.139 | 0.252 | 11 | 0.145 | 0.104 | 0.187 | 13 |
| Haiti | 2,090 | 0.410 | 8 | 0.256 | 0.207 | 0.305 | 8 | 0.248 | 0.200 | 0.296 | 7 |
| India | 76,079 | 0.632 | 20 | 0.070 | 0.062 | 0.079 | 29 | 0.065 | 0.057 | 0.073 | 23 |
| Indonesia | 6,043 | 0.680 | 23 | 0.112 | 0.081 | 0.143 | 21 | 0.098 | 0.071 | 0.125 | 17 |
| Jordan | 3,558 | 0.808 | 35 | 0.047 | −0.001 | 0.095 | 38 | 0.029 | −0.001 | 0.059 | 38 |
| Kenya | 6,940 | 0.716 | 25 | 0.110 | 0.080 | 0.140 | 23 | 0.089 | 0.065 | 0.114 | 18 |
| Lesotho | 1,029 | 0.687 | 24 | 0.079 | 0.003 | 0.155 | 27 | 0.068 | 0.003 | 0.133 | 21 |
| Liberia | 1,771 | 0.473 | 12 | 0.065 | 0.011 | 0.119 | 33 | 0.065 | 0.011 | 0.119 | 24 |
| Madagascar | 4,006 | 0.487 | 13 | 0.238 | 0.203 | 0.273 | 9 | 0.238 | 0.203 | 0.273 | 8 |
| Malawi | 5,623 | 0.725 | 26 | 0.066 | 0.032 | 0.100 | 31 | 0.053 | 0.026 | 0.080 | 27 |
| Maldives | 1,019 | 0.772 | 32 | 0.066 | −0.019 | 0.150 | 32 | 0.046 | −0.013 | 0.106 | 31 |
| Mali | 3,224 | 0.400 | 6 | 0.050 | 0.009 | 0.091 | 36 | 0.048 | 0.009 | 0.087 | 30 |
| Mauritania | 3,745 | 0.364 | 5 | −0.053 | −0.092 | −0.015 | 35 | −0.049 | −0.085 | −0.014 | 29 |
| Myanmar | 1,556 | 0.613 | 19 | 0.276 | 0.218 | 0.333 | 6 | 0.261 | 0.207 | 0.316 | 5 |
| Nepal | 1,710 | 0.793 | 33 | 0.026 | −0.042 | 0.093 | 40 | 0.017 | −0.028 | 0.061 | 41 |
| Nigeria | 10,212 | 0.295 | 4 | 0.380 | 0.357 | 0.404 | 2 | 0.316 | 0.297 | 0.336 | 2 |
| Pakistan | 4,075 | 0.675 | 21 | 0.332 | 0.295 | 0.368 | 3 | 0.291 | 0.259 | 0.323 | 3 |
| Philippines | 3,523 | 0.676 | 22 | 0.210 | 0.169 | 0.250 | 10 | 0.184 | 0.148 | 0.219 | 11 |
| Rwanda | 2,729 | 0.948 | 43 | 0.128 | 0.031 | 0.226 | 19 | 0.025 | 0.006 | 0.044 | 39 |
| Senegal | 2,103 | 0.746 | 29 | 0.170 | 0.113 | 0.226 | 14 | 0.129 | 0.086 | 0.171 | 16 |
| Sierra Leone | 3,030 | 0.541 | 16 | −0.035 | −0.076 | 0.007 | 39 | −0.034 | −0.075 | 0.007 | 36 |
| South Africa | 1,176 | 0.596 | 18 | −0.047 | −0.114 | 0.020 | 37 | −0.045 | −0.110 | 0.019 | 32 |
| Tajikistan | 2,190 | 0.821 | 36 | −0.068 | −0.130 | −0.007 | 30 | −0.040 | −0.076 | −0.004 | 35 |
| Tanzania | 3,484 | 0.744 | 28 | 0.173 | 0.129 | 0.216 | 13 | 0.132 | 0.098 | 0.165 | 14 |
| The Gambia | 2,649 | 0.831 | 38 | −0.053 | −0.112 | 0.005 | 34 | −0.030 | −0.063 | 0.003 | 37 |
| Timor-Leste | 2,416 | 0.454 | 11 | 0.169 | 0.123 | 0.215 | 15 | 0.168 | 0.122 | 0.213 | 12 |
| Uganda | 5,127 | 0.539 | 15 | −0.021 | −0.053 | 0.011 | 41 | −0.021 | −0.052 | 0.011 | 40 |
| Zambia | 3,323 | 0.733 | 27 | 0.077 | 0.033 | 0.121 | 28 | 0.060 | 0.026 | 0.095 | 26 |
| Zimbabwe | 2,009 | 0.761 | 31 | 0.110 | 0.051 | 0.169 | 22 | 0.080 | 0.037 | 0.123 | 20 |

The lower and upper bounds refer to the 95% confidence intervals of W and E. Countries are ranked from the worst performing (i.e., lowest vaccination rate or highest magnitude of inequality) to the best performing. Sampling weights were applied in all calculations.

FIC, full immunization coverage.

countries with the lowest FIC (Chad, Guinea, Angola, Nigeria, and Afghanistan) also display the widest disparities across administrative regions (S6 Table).

The LISA analysis, which aimed at identifying clusters with consistently low or high values for each country, reveals that FIC is low throughout Africa, particularly in areas of Nigeria, Cameroon, and Tanzania, as well as in north-eastern India and south-western Afghanistan (Fig 1). Clusters of high levels of FIC can be distinguished in some countries in west Africa (parts of Senegal, Nigeria, and Chad) and central and southern India, with a high degree of statistical confidence.

We use Wagstaff's (W) and Erreygers' (E) indices to measure socioeconomic inequalities in children's full immunization status. At the national level, the 2 indices result in a similar but not identical ranking of countries by level of socioeconomic inequality (Table 1). In 30 of the 43 countries, we detected pro-rich inequalities at a high level of statistical significance, while in 3 countries (Tajikistan, Mauritania, and Armenia), we detected pro-poor inequalities at a high level of statistical significance. In these 3 countries, better-off groups of the population are less likely to be fully vaccinated. Most countries display a low degree of socioeconomic inequality (absolute values of W and E between 0 and 0.15; Table 1). However, 3 countries stand out with a more considerable degree of inequality—Angola, Nigeria, and Pakistan—with both W and E values of 0.3 or higher, implying that vaccination there is concentrated among the wealthier groups.

At the subnational level, the magnitude of socioeconomic inequality varies more substantially (Figs 2 and 3 and S5 Table). India shows the widest heterogeneity among administrative divisions, with W values ranging from 0 in the best-performing district to 0.87 in the worst-performing district and E values ranging from 0 to 0.68. Wide subnational disparities can be seen in Indonesia, Ethiopia, and Myanmar as well, with a difference between the best and the worst performing administrative divisions of 0.5 or higher for both W and E. High degree of pro-rich inequality is found across Africa, particularly throughout Angola, Nigeria, Ethiopia, and Madagascar, as well as throughout Afghanistan and Pakistan, and these estimates are statistically significant at 95 percent confidence level (see S3 Fig).

Spatial clusters with high degrees of socioeconomic inequalities in children's full immunization status are identified via LISA analysis (Figs 2 and 3). Such clusters can be distinguished with a high degree of statistical confidence in a few regions in Africa, mainly in areas of Zambia, Madagascar, and Benin, as well as in Indonesia and central India.

S4 and S5 Figs show the spatial intersection between the full vaccination coverage and the degree of socioeconomic inequality at the subnational level. Distinct spatial patterns are revealed with regions of Nigeria, Chad, Cameroon, Guinea, Madagascar, Angola, and Ethiopia characterized by both low vaccine coverage and a substantial degree of socioeconomic inequality in child immunization, which implies a double disadvantage for poor households living there (dark red and orange areas in S4 and S5 Figs). Such patterns can be seen in several regions in Pakistan, Afghanistan, and Haiti as well. In contrast, a high level of vaccine coverage and a low degree of socioeconomic inequality is observed in most of southern and eastern Africa (with Ethiopia being a notable exception), throughout India, and across Nepal, Bangladesh, and Tajikistan.

Fig 4 shows the type of vaccines or vaccine combinations that are most often lacking across all countries included in the analysis. We can see that incomplete immunization is most often due to children not receiving the MCV—and the OPV, which requires 3 doses to complete the immunization cycle. BCG, which requires a single dose and is usually administered soon after birth, was the least likely to be missed.

A substantial number of children aged 15 to 35 months lack all 4 essential vaccines included in the EPI schedule (10.5 million). The consistent lack of all 4 vaccinations reveals a large

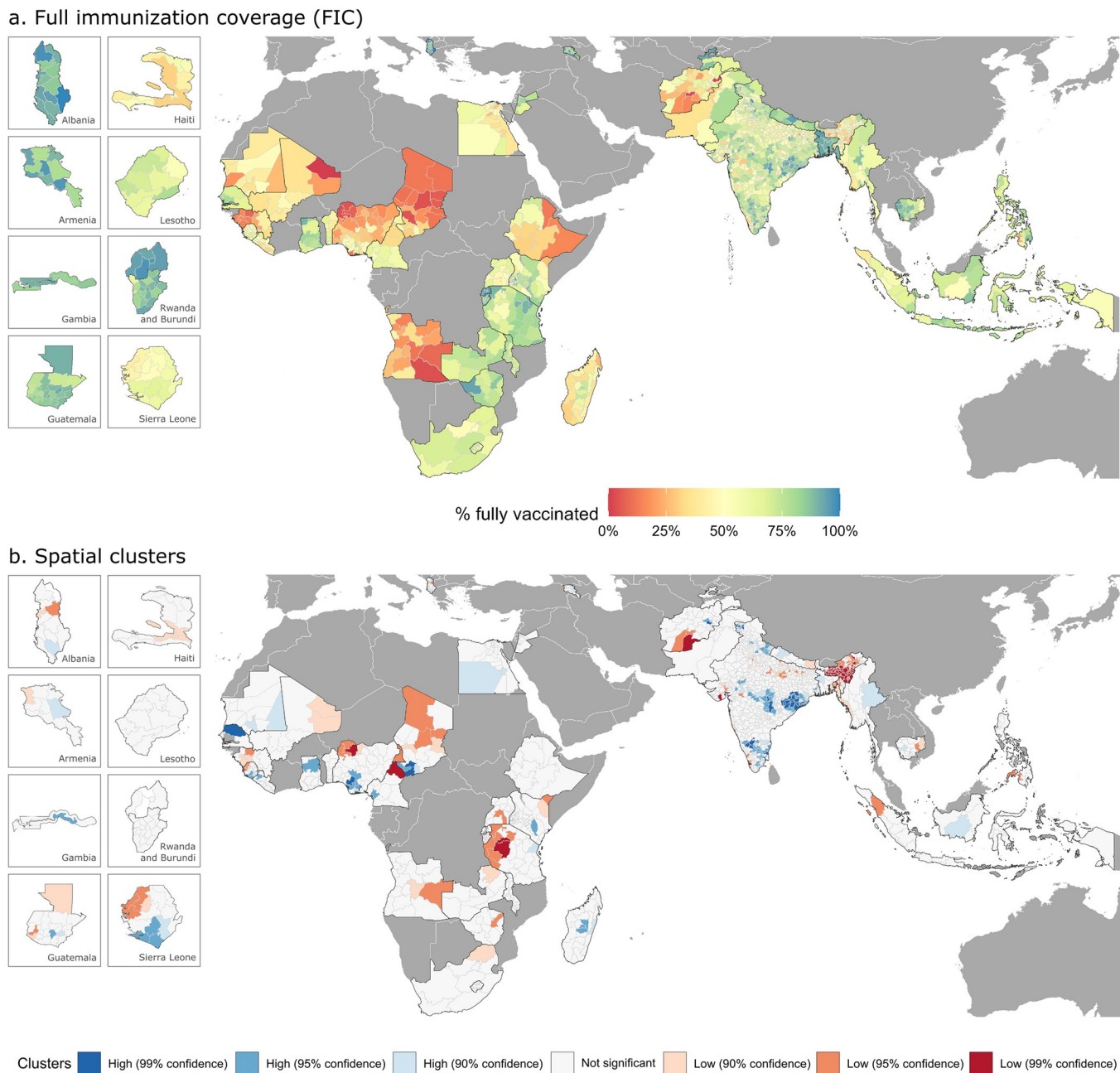

**Fig 1.** Subnational estimates of FIC (a) and spatial clusters of administrative regions with high (blue colors) and low (red colors) values of FIC (b). Spatial boundaries were retrieved from Natural Earth (https://www.naturalearthdata.com/) using the "rnaturalearth" package (https://github.com/ropenscilabs/rnaturalearth). FIC, full immunization coverage.

immunization gap, possibly due to poor access to immunization and other health services, and is particularly the case in sub-Saharan Africa, Afghanistan, parts of Southeast Asia, and Haiti.

We checked the sensitivity of the above results by estimating immunization rates and levels of socioeconomic inequality for children between 24 and 35 months of age. The results are available in S3 and S4 Tables and are comparable to the findings for children aged 15 to 35 months presented above.

**Table 2. Theil indices and their components for coverage of BCG vaccine, DTP vaccine, OPV, MCV, FIC, Wagstaff's (W), and Erreygers' (E) indices of inequality.**
Results for individual countries are available in S6 Table.

|  | BCG | DTP | OPV | MCV | FIC | W | E |
|---|---|---|---|---|---|---|---|
| Theil index | 0.011 | 0.029 | 0.028 | 0.028 | 0.059 | 0.311 | 0.315 |
| Within | 0.006 | 0.011 | 0.011 | 0.018 | 0.032 | 0.271 | 0.277 |
| Between | 0.006 | 0.018 | 0.016 | 0.010 | 0.028 | 0.040 | 0.038 |

BCG, bacille Calmette–Guerin; DTP, diphtheria–tetanus–pertussis; FIC, full immunization coverage; MCV, measles-containing vaccine; OPV, oral polio vaccine.

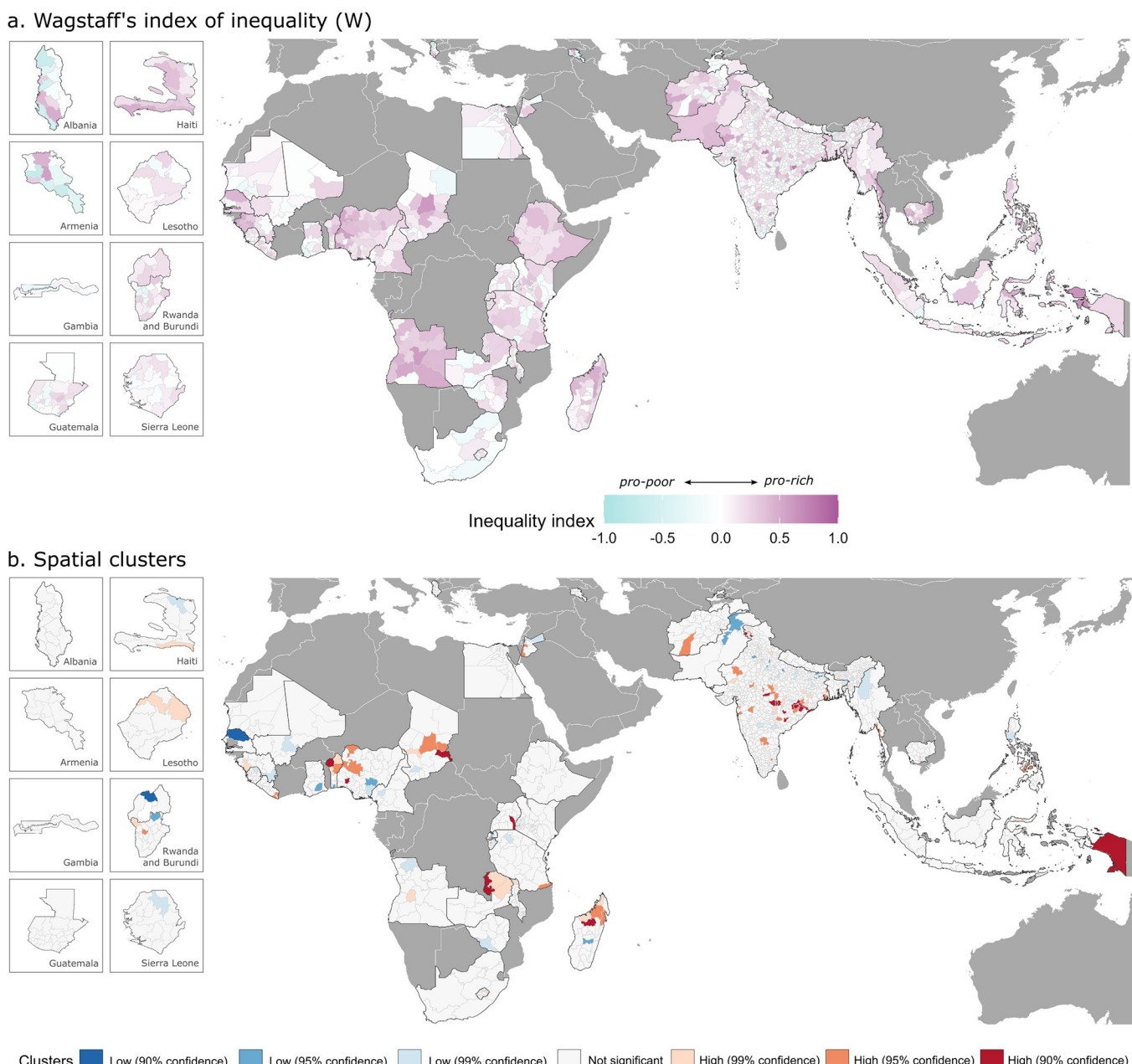

**Fig 2.** Subnational estimates of Wagstaff's index (W) of socioeconomic inequality (a) and spatial clusters of administrative regions with high (red colors) and low (blue colors) degrees of inequality (b). Spatial boundaries were retrieved from Natural Earth (https://www.naturalearthdata.com/) using the "rnaturalearth" package (https://github.com/ropenscilabs/rnaturalearth).

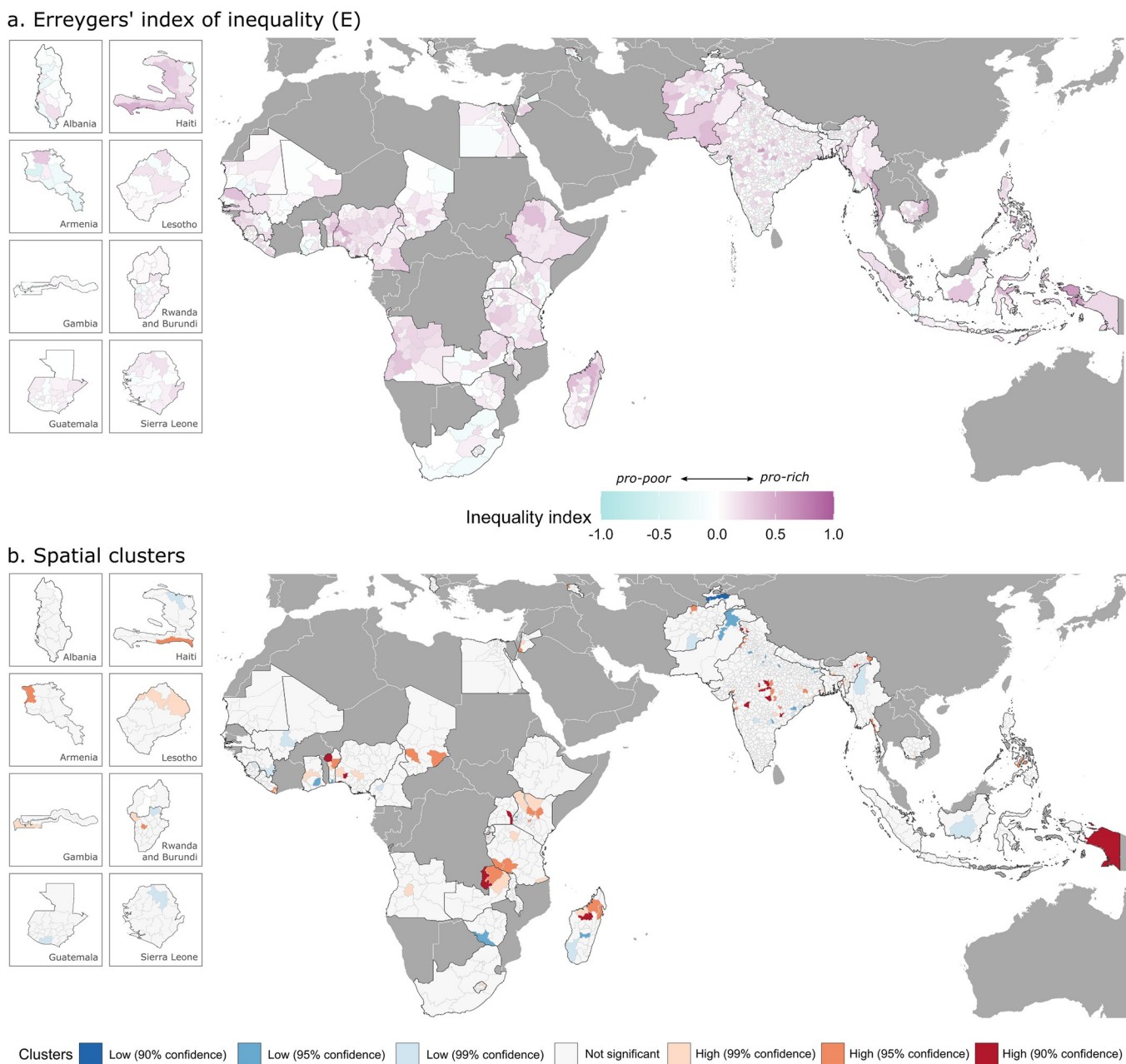

**Fig 3.** Subnational estimates of Erreygers' index (E) of socioeconomic inequality (a) and spatial clusters of administrative regions with high (red colors) and low (blue colors) degrees of inequality (b). Spatial boundaries were retrieved from Natural Earth (https://www.naturalearthdata.com/) using the "rnaturalearth" package (https://github.com/ropenscilabs/rnaturalearth).

## Discussion

Global efforts, particularly WHO's Expanded Program on Immunization [27], Gavi, the Vaccine Alliance [50], and more recently the Global Vaccine Action Plan [6], have been largely successful in delivering essential vaccines to poor countries where child mortality from common infectious illnesses has declined as a result [5,51]. However, progress in child immunization has started to slow down [6,7] and even reverse in recent years [12], particularly in sub-Saharan Africa and South Asia where common infectious diseases remain a major health issue. The Coronavirus Disease 2019 (COVID-19) pandemic has exposed the modern challenges

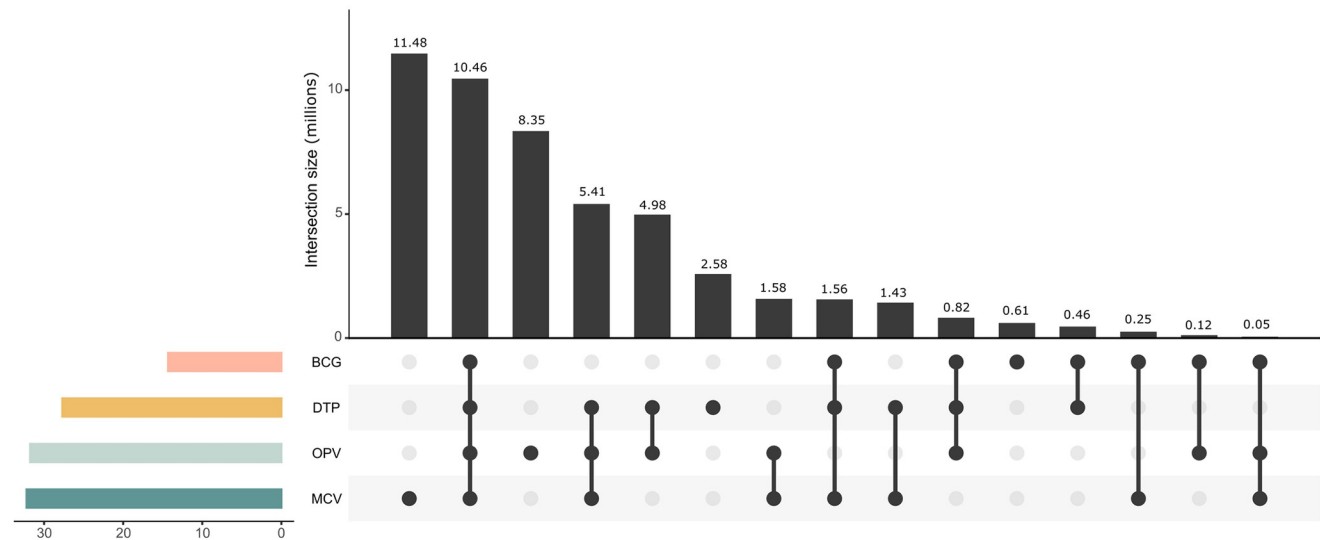

a. Intersecting sets of missed vaccinations across all 43 countries

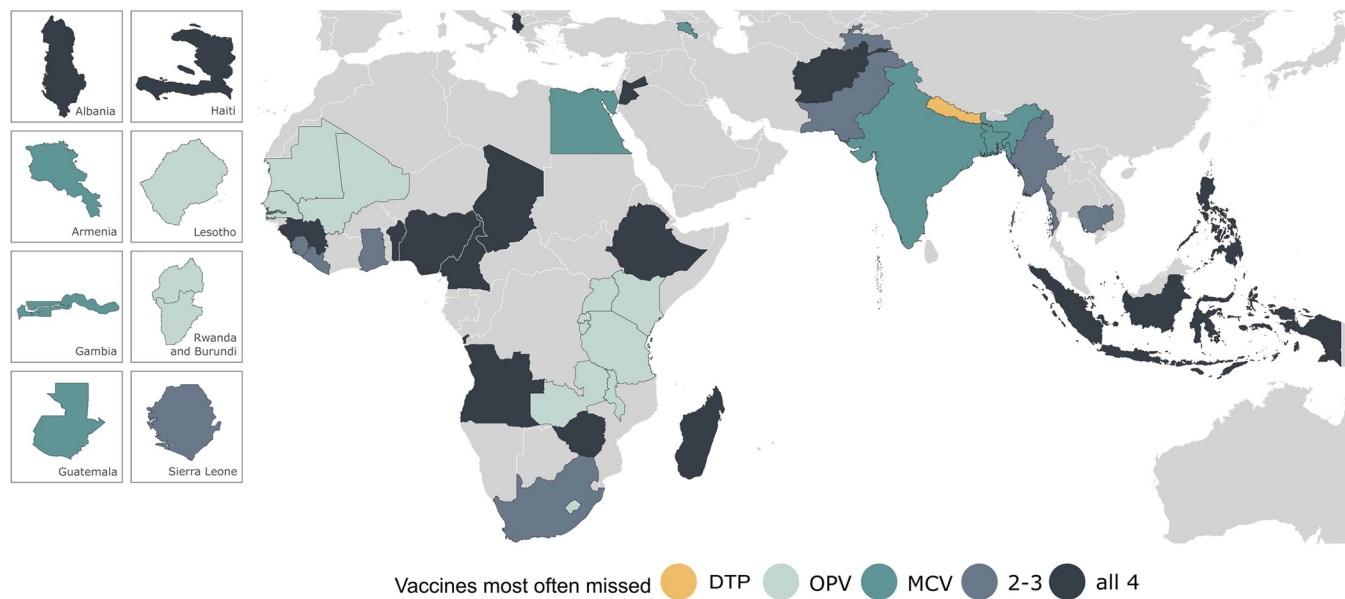

b. Types of vaccines most often missed per country

**Fig 4.** Intersecting sets of missed vaccinations among children aged 15 to 35 months across all 43 countries (a) and type of vaccines most often missed per country (b). In panel (a), the horizontal bars indicate the number of children aged 15 to 35 months that have missed each vaccination, and the dots and vertical bars indicate the combinations of vaccinations missed. Detailed country-level estimates are provided in S6 Fig. Note that multiple doses of DTP and OPV are needed to reach complete immunization. Spatial boundaries were retrieved from Natural Earth (https://www.naturalearthdata.com/) using the "rnaturalearth" package (https://github.com/ropenscilabs/rnaturalearth). BCG, bacille Calmette–Guerin vaccine; DTP, diphtheria–tetanus–pertussis vaccine; MCV, measles-containing vaccine; OPV, oral polio vaccine.

faced by poor countries in accessing life-saving vaccines. Moreover, LMICs are consistently left behind when it comes to the rapid evolution of global immunization databases, which were developed in response to the COVID-19 pandemic and provide up-to-date information at a fine temporal and spatial resolution [52]. Subnational heterogeneities in essential immunization remain understudied in LMICs, which presents a major barrier to the design and implementation of context-specific interventions there [53].

Providing universal access to vaccines by 2030 is one of the key objectives of Sustainable Development Goal 3 [54]. To make sure that progress remains on track, it is imperative that communities with a high share of under- and unvaccinated individuals are identified and barriers to receiving life-saving immunization addressed. However, vaccination coverage is usually reported at the national level, which can conceal large subnational heterogeneities as demonstrated in this study. Using data from 43 low- to middle-income countries, we demonstrate the presence of both within-country and between-country disparities in essential childhood immunization. Furthermore, we identify clusters of administrative regions characterized by low vaccine coverage and a high degree of socioeconomic inequality in essential childhood immunization.

Our findings reveal large gaps in child immunization throughout Africa and in South Asia, which demonstrates the need to reinforce immunization efforts in these regions. Some of the lowest vaccination rates are observed in areas of Angola, Chad, Nigeria, Guinea, Mali, and Afghanistan, where less than 10% of children are fully immunized. Furthermore, we find that most children in these countries lack all 4 basic vaccines included in WHO's EPI schedule (BCG, DTP, OPV, and MCV), which implies generally poor access to immunization and health services there. Closing the vaccination gap in these locations may prove particularly difficult.

Low vaccination rates also seem to coincide with a high degree of socioeconomic inequality in children's immunization status. Across most countries, we find that children from poorer households are less likely to be fully immunized. Our results are in line with previous studies, which find pro-rich inequalities in FIC using a variety of inequality metrics [14–16]. The combination of both low vaccine coverage and high socioeconomic inequality in essential immunization in certain regions is particularly concerning since poor households are the ones that are most likely to be living in unsanitary conditions [55], experience food insecurity [56], and lack access to health services [57], all of which contribute to high child morbidity and mortality from infectious diseases [58]. Children living in such vulnerable situations should be prioritized in immunization programs.

Interestingly, in a few countries, we found that children from wealthier households were less likely to be fully immunized. This was particularly the case in Armenia, Tajikistan, and Mauritania. Patterns of increasing pro-poor inequality in child immunization across some LMICs have been reported in the literature before [14,16]. The emergence of vaccine hesitancy, which is usually observed in more economically developed countries [59,60], could explain this phenomenon [61]. Various factors may influence vaccine hesitancy, such as parents' perception of the risks and benefits of child immunization. However, knowledge about vaccine hesitancy and its impact on vaccine uptake in low-income countries is still limited [61,62]. A better understanding of these trends in LMICs is needed to ensure the success of future vaccination campaigns.

This study has several limitations. The vaccination information was verified via vaccination cards for 67% of children in the sample. For the rest, this information was based on the mother's recall, which is subject to recall bias. The validity of relying on parental reporting of immunization is shown to vary in the context of LMICs [63–65]. Another limitation is that we have not been able to distinguish between vaccines administered as part of routine immunization services versus immunization campaigns due to the lack of such data. The effectiveness of immunization campaigns as compared to routine immunization services is unclear—some evidence shows that such campaigns reduce health inequalities [66], while other research shows limited effectiveness in the long term [67]. Moreover, the period of data collection ranges from 2014 and 2021, which may affect the comparability of results across countries. In a few countries included in the analysis, survey data were collected after the onset of the COVID-19

pandemic in early 2020. In most countries, however, the data were collected before 2020, which implies that the disruption of vaccination efforts due to the COVID-19 pandemic will not be reflected there. Moreover, our results are not representative of all LMICs. At the subnational level, vaccination coverage is estimated at different administrative levels (e.g., states, provinces, or districts), depending on the spatial scales at which the DHS surveys are representative, which limits the detection and comparison of spatial clusters. Another limitation concerns the small sample sizes for some administrative regions, which may result in imprecise estimates. We have provided 95% confidence intervals for the inequality indices to account for that. The wealth index, which is used to determine the socioeconomic position of households, is a relative measure of poverty and results may be different with respect to absolute measures of poverty.

We highlight certain research directions that can be explored in the future. Finer spatial resolution maps can be produced using state-of-the-art statistical tools and remote sensing data [13]. In the recent literature, advanced geostatistical techniques have been used to generate subnational estimates for various development indicators at a high spatial resolution, i.e., gridded pixel level [13,68,69]. Such downscaling methods present an opportunity for identifying under-vaccinated communities more precisely and should be explored in future research.

While in this study we have focused on spatial and socioeconomic inequalities in FIC, other forms of inequalities have also been found in the literature. There is evidence of large disparities in immunization coverage with respect to ethnicity [70], area of residence [16], female empowerment [71], and overall access to primary healthcare services [72], among other factors [8]. Our study complements these findings and emphasizes the importance of monitoring inequalities across multiple dimensions.

Improving vaccine coverage in LMICs will not only be critical for reducing the enormous burden of infectious illnesses in these places but it can also facilitate progress toward other development objectives. Continuous exposure to infections can impair children's long-term growth and development through its complex interaction with malnutrition [73–75]. Moreover, the presence of infections in childhood has been associated with missed school days [76,77] and lower cognitive performance [78], which can keep disadvantaged children in a poverty trap. Vaccination can also play a key role in reducing the burden of antimicrobial resistance. A recent study estimated that 2 vaccines—pneumococcal conjugate vaccines and live attenuated rotavirus vaccines—prevent 23.8 million and 13.6 million episodes of antibiotic-treated illnesses annually among children under 5 in LMICs [79]. Achieving universal immunization will be central to the success of a number of development priorities [80,81].

As new vaccines become available, it is important to ensure that they are equally distributed both between and within countries. The ongoing experience with the COVID-19 pandemic and the ensuing vaccine rollout has laid bare the structural inequities in access to vaccines globally that yet need to be addressed. By early 2022, 72% of all COVID-19 vaccine doses had been administered in high- and upper-middle-income countries and only 0.9% of all doses had been administered in low-income countries [82]. As demonstrated in this study, such inequities can be seen with respect to essential childhood immunization as well. Moreover, the hard-won gains in essential immunization achieved in the past 5 decades risk being undone due to the COVID-19 pandemic and the reported disruption in immunization programs across the world [83,84].

The accumulation of lacking vaccines in poor countries, as demonstrated in this study, is an indication of structural barriers with regard to vaccine access. While those populations that are easy to reach have generally been well served, reaching the less-accessible populations, including those in remote rural and conflict affected areas and the urban poor, has proven challenging [6]. Within-country heterogeneities in essential immunization remain understudied, which presents a major barrier to the design and implementation of context-specific

interventions [53]. Addressing existing barriers to vaccination will be beneficial for ongoing COVID-19 vaccination efforts and for limiting the burden associated with the pandemic and the rapid virus mutation. Securing vaccines for poor countries and under-vaccinated communities within these countries needs to become a greater priority to ensure that the health gap between rich and poor nations does not continue to grow.

## Supporting information

**S1 Table. STROBE Statement—Checklist of items that should be included in reports of cross-sectional studies.**
(PDF)

**S2 Table. Overview of DHS surveys included in the analysis.** The subnational units are states, regions, or districts, depending on the survey. The sample includes children between 15 and 35 months of age.
(PDF)

**S3 Table. National estimates of vaccine coverage for BCG, DTP, OPV, and MCV among children in indicated age groups.** Sampling weights were applied in all calculations.
(PDF)

**S4 Table. National estimates of FIC, Wagstaff's index of inequality (W), and Erreygers' index of inequality (E) among children between 24 and 35 months of age.** The lower and upper bounds refer to the 95% confidence intervals of W and E. Countries are ranked from the worst performing (i.e., lowest vaccination rate or highest magnitude of inequality) to the best performing. Sampling weights were applied in all calculations.
(PDF)

**S5 Table. Subnational estimates of FIC, Wagstaff's (W), and Erreygers' (E) indices of inequality.** The lower and upper bounds refer to the 95% confidence intervals of W and E. Sampling weights were applied in all calculations.
(PDF)

**S6 Table. Theil index of inequality by country for individual vaccinations, FIC, and Wagstaff's (W) and Erreygers' (E) indices of inequality.**
(PDF)

**S1 Fig. Location of PSUs for which geocoordinates (latitude and longitude) are available in DHS.** No geocoordinate information is available for PSUs in Afghanistan, Maldives, Mauritania, and Indonesia. Spatial boundaries were retrieved from Natural Earth (https://www.naturalearthdata.com/) using the "rnaturalearth" package (https://github.com/ropenscilabs/rnaturalearth).
(PDF)

**S2 Fig. A concentration curve based on DHS survey data from Nigeria for children aged 15 to 35 months.** The $y$-axis shows the cumulative share of children who are fully immunized, and the $x$-axis shows the cumulative share of children ranked by wealth index from the poorest to the richest. The green 45˚ line represents a state of perfect equality.
(PDF)

**S3 Fig. Subnational estimates of Wagstaff's (a) and Errgeyers' (b) indices of inequality statistically significant at a 95 percent level.** Spatial boundaries were retrieved from Natural Earth (https://www.naturalearthdata.com/) using the "rnaturalearth" package (https://github.com/ropenscilabs/rnaturalearth).
(PDF)

**S4 Fig. Bivariate map showing the intersection between FIC and Wagstaff's (W) index of inequality.** Spatial boundaries were retrieved from Natural Earth (https://www.naturalearthdata.com/) using the "rnaturalearth" package (https://github.com/ropenscilabs/rnaturalearth).
(PDF)

**S5 Fig. Bivariate map showing the intersection between FIC and Erreygers' (E) index of inequality.** Spatial boundaries were retrieved from Natural Earth (https://www.naturalearthdata.com/) using the "rnaturalearth" package (https://github.com/ropenscilabs/rnaturalearth).
(PDF)

**S6 Fig. Intersecting sets of missed vaccinations for children aged 15 to 35 months for 43 countries.** The black dots represent vaccine combinations and the bars represent the number of missed vaccinations for each vaccine combination. Note that multiple doses of DTP and OPV are needed to reach full immunization; therefore, the presented estimates do not refer to the number of missed doses but complete vaccinations. If a child is missing 2 or more doses of a specific vaccine, the child will be counted only once.
(PDF)

## Author Contributions

**Conceptualization:** Anna Dimitrova, Gabriel Carrasco-Escobar, Robin Richardson, Tarik Benmarhnia.

**Data curation:** Anna Dimitrova, Gabriel Carrasco-Escobar.

**Formal analysis:** Anna Dimitrova, Gabriel Carrasco-Escobar, Tarik Benmarhnia.

**Investigation:** Gabriel Carrasco-Escobar, Tarik Benmarhnia.

**Methodology:** Anna Dimitrova, Gabriel Carrasco-Escobar, Tarik Benmarhnia.

**Project administration:** Anna Dimitrova, Tarik Benmarhnia.

**Supervision:** Robin Richardson, Tarik Benmarhnia.

**Validation:** Anna Dimitrova, Gabriel Carrasco-Escobar, Robin Richardson, Tarik Benmarhnia.

**Visualization:** Anna Dimitrova, Gabriel Carrasco-Escobar.

**Writing – original draft:** Anna Dimitrova, Gabriel Carrasco-Escobar, Robin Richardson, Tarik Benmarhnia.

**Writing – review & editing:** Anna Dimitrova, Gabriel Carrasco-Escobar, Robin Richardson, Tarik Benmarhnia.

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
