## [Editor Report · Decision Letter 0]

31 May 2022

Dear Dr Dimitrova, 

Thank you for submitting your manuscript entitled "Subnational inequalities in essential childhood immunization across 40 low- and middle-income countries" for consideration by PLOS Medicine.

Your manuscript has now been evaluated by the PLOS Medicine editorial staff and I am writing to let you know that we would like to send your submission out for external peer review.

Please re-submit your manuscript within two working days, i.e. by Jun 02 2022 11:59PM.

Kind regards,

Beryne Odeny

PLOS Medicine

---

## [Decision Letter · Decision Letter 1]

20 Jul 2022

Dear Dr. Dimitrova,

Thank you very much for submitting your manuscript "Subnational inequalities in essential childhood immunization across 40 low- and middle-income countries" (PMEDICINE-D-22-01743R1) for consideration at PLOS Medicine. 

[LINK]

In light of these reviews, I am afraid that we will not be able to accept the manuscript for publication in the journal in its current form, but we would like to consider a revised version that addresses the reviewers' and editors' comments. Obviously we cannot make any decision about publication until we have seen the revised manuscript and your response, and we plan to seek re-review by one or more of the reviewers. 

We expect to receive your revised manuscript by Aug 10 2022 11:59PM. Please email us (plosmedicine@plos.org) if you have any questions or concerns.

We look forward to receiving your revised manuscript. 

Sincerely,

Beryne Odeny, 

PLOS Medicine

plosmedicine.org

Comments from the Academic Editor:

 My main concern is lack of representativeness and power to support inference of the small area units used for spatial analysis, as pointed out by several reviewers as well. These are further divided by examining inequalities within those small regions (i.e., even fewer individuals per wealth group). The uncertainty estimates can only be very high here and this should be more explicitly discussed in limitations--as well as in sensitivity analyses, say dropping the smallest units. Some of the maps are not effective given the relatively few countries/areas in the analysis--see Figure 4. Otherwise, this is a fine paper though not entirely novel.

1) Please revise your title according to PLOS Medicine's style. Your title must be nondeclarative and not a question. It should begin with main concept if possible. Please place the study design (for example, "Spatial cluster analysis”) in the subtitle (ie, after a colon). 

2) Abstract

a) Please structure your abstract using the PLOS Medicine headings (Background, Methods and Findings, Conclusions).

b) Please include the actual amounts of relevant outcomes. Please provide the key estimates and confidence intervals in the Methods and Finding section

c) Please include the important dependent variables that are adjusted for in the analyses.

d) In the last sentence of the Abstract Methods and Findings section, please describe the main limitation(s) of the study's methodology.

3) Author summary - At this stage, we ask that you include a short, non-technical Author Summary of your research to make findings accessible to a wide audience that includes both scientists and non-scientists. The Author Summary should immediately follow the Abstract in your revised manuscript. This text is subject to editorial change and should be distinct from the scientific abstract. Please see our author guidelines for more information: https://journals.plos.org/plosmedicine/s/revising-your-manuscript#loc-author-summary.

4) For streamlining, the last paragraph of the Background can be moved to the discussion section as you begin discussing the implications of the findings of your study

5) As alluded to by most reviewers, the novelty aspect of this paper needs further elaboration and justification.

6) Please remove the “Role of the funding source” statement from the Methods. In the event of publication, this information will be published as metadata based on your responses to the submission form.

7) Please ensure that the study is reported according to the STROBE guideline or similar, and include the completed STROBE checklist as Supporting Information. When completing the checklist, please quote section and paragraph numbers rather than page numbers. Please add the following statement, or similar, to the Methods: "This study is reported as per the Strengthening the Reporting of Observational Studies in Epidemiology (STROBE) guideline (S1 Checklist)."

8) Please temper claims of primacy in abstract and main text (e.g., “we provide the first global analysis”) by stating, "to our knowledge" or something similar.

9) For your Tables and figures, please do the following:

a) In figure 1 and 3, please indicate in the figure caption the meaning of the bars and whiskers

b) Please define the abbreviations such as MCV, DTP, OPV, BCG

c) Please confirm that the appropriate usage rights apply to the use of this map. Please see our guidelines for map images: https://journals.plos.org/plosmedicine/s/figures#loc-maps

10) References:

a) Please select the PLOS Medicine reference style in your citation manager. In-text reference call outs should be presented as follows noting the absence of spaces within the square brackets, e.g., “…population [1,2].”

b) Please ensure that journal name abbreviations match those found in the National Center for Biotechnology Information (NCBI) databases and are appropriately formatted and capitalized. https://journals.plos.org/plosmedicine/s/submission-guidelines#loc-references.

c) Please ensure that weblinks are current and accessible to date.

Comments from the reviewers:

Reviewer #1: RE: Subnational inequalities in essential childhood immunization across 40 low- and middle-income countries

The paper used the most recent available Demographic Health Surveys (DHS) to assess spatial and socioeconomic inequalities in child immunization in 40 low- and middle-income countries (LMICs). The authors provide national and subnational statistics about vaccination rates for the countries included in the study. Additionally, the authors used the concentration index approach to report wealth-related inequalities in vaccination uptake in the countries. To better present the finding, the paper uses special illustration techniques to better present areas with higher/lower vaccination and socioeconomic inequalities in LMICs. 

Comments:

1- Although the paper addresses an important health issue, the literature review provided in the study is not comprehensive. The authors claim that this study "provide the first global analysis of spatial and socioeconomic inequalities in essential vaccine coverage." The paper also states that "no global assessment has been carried out to date based on the full EPI schedule for young children". These statements are incorrect as there are recent studies that measured and even explained socioeconomic inequalities in vaccination uptake in LMICs. For example, see https://www.ncbi.nlm.nih.gov/pmc/articles/PMC5096343/

https://jech.bmj.com/content/72/8/719
https://www.sciencedirect.com/science/article/pii/S0749379720303950

The paper does not mention any of these papers. The authors need to conduct a comprehensive literature review and highlight the main contribution of their study given the relevant studies on this topic. 

2- The paper does poorly when it explains its data source. There should be further information about the DHS. For example, one may wonder what the response rate of DHS for each country is. How the data have been collected?

3- It is unclear whether the paper considered the survey design in their estimation. For example, did the authors used sampling weight in the analyses?

4- If the MCV should be administrated by 9 months, why does the study restrict its sample to the children 12-35 month?

5- The existing literature on socioeconomic inequalities in health suggests using the Wagstaff index and the Erreygers Index when we measure socioeconomic inequalities for binary health outcomes. One may wonder why the paper only used the Wagstaff index in their study. 

6- The authors can provide further information on variation in vaccination rates using summary inequality measures such as the Theil index (T), between-group variance (BGV).

7- On page 11, what range of the C was used in their following statement, "low versus high socioeconomic inequalities in childhood immunization status"? 

Reviewer #2: Overview: In this manuscript, the authors conduct a series of secondary analyses of DHS data to explore patterns of full immunization coverage and their intersections with socioeconomic status in the most recent DHS surveys for 40 low- and middle-income countries. Similar data for some of the results presented here are already available from standard DHS reports and resources online; for instance, subnational full-vaccination coverage among 12-23 month-olds is a standard component of DHS reports; here, the authors use 12-35 months as the age group of analysis instead. In this manuscript, however, the authors compile these indicators across multiple countries for easier comparison and use standard geospatial analytic techniques to identify clusters of high and low coverage. Multiple manuscripts have applied these sorts of methods in the past, but usually for single countries or larger groups of countries; the novelty in this part of the analysis comes primarily from its application across a relatively large number of countries. 

In addition, however, the authors also look at subnational patterns of socioeconomic inequalities in full vaccinated coverage. This interesting analysis provides insight into the ways in which two different categories of vaccination inequality (spatial and socioeconomic) intersect within countries. They also analyze the specific patterns of missing vaccines within and across countries - i.e. if a child is not fully vaccinated, which vaccine(s) are most likely to be incomplete? In all, the manuscript is clearly written and provides several useful additions to the accumulated body of literature that uses DHS data to analyze patterns of vaccination coverage. I have a few major comments (below) and several minor comments for the authors' consideration, provided in the hope that they will further strengthen the manuscript. 

Major comments:

1. In general, the manuscript would greatly benefit from more attention given to the uncertainty inherent in these analyses. While DHS surveys are designed to be representative at these subnational spatial scales, smaller sample sizes generally mean that the uncertainty around subnational estimates generated using traditional survey statistics for these subnational units may be wider. Did the authors use only mean estimates of coverage in all of these analyses, or was uncertainty accounted for in some way? Could the authors provide 95% CI or UI ranges for figures cited, i.e. in lines 232-236 and elsewhere? In particular, it would be useful to know how much uncertainty there is in the subnational estimates of full immunization rates and concentration index values. 

2. The manuscript also should contain a thorough discussion of the limitations of this analysis and the results. Unless I am missing it, this is not currently present in the discussion section. There are a few places where caveats are given - for instance, in lines 149-152, the authors note the (rather major) assumption that coverage hasn't changed since the last DHS survey for each country. The fact that the included DHS surveys took place over a range of 5 years, and that some of these surveys were not conducted almost 8 years ago, limits both comparability of the results and inference about the global findings. This and other limitations - a few of which I have remarked on below, but the authors likely would want to comment on others - should be discussed in some detail. 

Minor comments:

1. Line 104-106: The authors used a standard EPI schedule to set the 12-35 month time frame for vaccination. This makes sense for BCG, DTP and OPV, all of which are recommended for administration in the first few weeks or months of life. The schedule for measles, however, is variable, depending on countries' measles epidemiology, etc. (https://immunizationdata.who.int/pages/schedule-by-disease/measles.html?ISO_3_CODE=&TARGETPOP_GENERAL=). How did the authors take this into account, as this is likely to affect comparability of results for children who are at the younger end of this age range? 

2. Also for MCV, were the authors able to distinguish between vaccines given via routine immunization services vs campaigns? This is often a challenge when working with DHS data, particularly in cases where one needs to rely on parental recall.

3. The authors focus mainly on geographic and socioeconomic inequalities in vaccination coverage in this analysis. Other analyses have focused on the use of individual-level survey data to better understand inequalities, although at aggregate rather than subnational scales. The authors may consider referencing some of these analyses to strengthen and expand their discussion of other potential forms of inequality in vaccination coverage, for instance https://pubmed.ncbi.nlm.nih.gov/35577393/, https://pubmed.ncbi.nlm.nih.gov/35356658/, and/or https://pubmed.ncbi.nlm.nih.gov/34805814/. 

4. A minor point, but I would suggest that the authors take some care with language describing the scale of the spatial heterogeneities in their analysis. The authors restrict the analysis to the subnational scales at which each DHS survey is representative using traditional survey statistics, i.e. the first administrative level (states, etc.) for some countries, the second administrative level (districts, etc.) for others. (As an aside, the different spatial scales are probably a limitation that is worth mentioning, as it does limit the comparability of some of the clustering and other analyses presented here). In some places, however, the authors refer to their results as exploring "fine-scale spatial heterogeneities". I might suggest that state-level spatial heterogeneities are not truly "fine-scale"; countries routinely use district-level administrative data to guide decision-making, and as the authors note there are a number of research groups (including DHS themselves, https://spatialdata.dhsprogram.com/modeled-surfaces/) who are using geostatistical modeling approaches to estimate gridded or lower-level administrative unit vaccine coverage indicators. The authors might consider ensuring that language describing the spatial scale of the analyses is clear throughout. 

Reviewer #3: See attachment

Michael Dewey

Reviewer #4: Review Notes

Overall this is a well-written manuscript, with clear descriptions of methodology and of the implications of their findings. The graphical presentation of results in particular is well-done and readily accessible to most readers. The demand for comprehensive sub-national data has grown through the era of COVID-19, and this manuscript has a role in ensuring that low and middle income countries (LMIC) have a place in such future data planning. As noted below, there are some methodological questions the authors need to resolve, such as whether the WHO schedule applied by the authors is actually congruent with individual national immunization schedules. Other questions include whether the selected age range (12 to 35 months) is nationally appropriate as compared to an older (24 to 35 month) age range. Also the authors need to address whether they have assessed on-time immunizations only, as opposed to immunizations given at any age prior to the 12-35 month individual end-point.

Specific Comments

Abstract

The claim that this is the first global study looking at sub-national gradients of immunization is somewhat off-target. The DHS survey covers a range of low to middle income countries, rather than the entire globe. There have been previous studies using DHS or MICS to map immunizations across multi-country regions, as the authors note. 

Introduction

Pg3Para2: The authors should spell-check their work here.

Pg3Para2: An extensive body of work exists identifying sub-national areas of low immunization. Such work however is usually on a single country basis, with various methodologies. The authors' text here should be modified to recognize that such work has and is proceeding- and what is needed is a coordinated approach across multiple countries as the authors present.

Pg3Para3: As in the introduction, the authors are over-selling their work here as the first global study. This statement should be more conditional, for participating LMICs, and within the context of the DHS.

Pg3Para3: The subclause "(with the finest spatial resolution…)" should be removed, as this belongs in the methodology discussion about mapping techniques.

Pg4Para3: This paragraph would be better placed in the Discussion section. It should also have a sentence regarding the rapid evolution of detailed global immunization databases around COVID, (Guidotti, 2022), and who LMIC are being left out of this evolution- definitely support for a niche program as the authors propose.

Study measures

Pg6Para2: The analysis is per the EPI schedule- however what proportion of LMICs exactly mimic this in their national schedules? MCV is listed at 9 months of age, but many LMIC list a minimum age of 12 months as is common in higher income nations. Similarly some national schedules call for BCG at birth, without provision for later catchup.

Pg6Para2: An issue with the current study is the inclusion of children from ages 12 to 35 month without assessing the relation between age and full immunization. Would the results appear different if limited to children age 24 to 35 months, as this would accommodate more local variation in immunization practice and some catchup opportunity?

Pg6Para2: The text suggests that immunizations were only counted if received by the EPI schedule due date- the authors should confirm if this is true, and if so then rerun results accounting for a longer period. Otherwise the language describing the study should be changed to reflect that this is an analysis of on-time immunization.

Pg6para3: The use of the Concentration Indexes is interesting here- I would suggest that the authors also present charts showing the relation between the probability of immunization against the underlying Wealth Index to provide more context to readers.

Spatial cluster analysis

Pg8Para2: The authors adjust for FDR (false discovery rate) using an approach from Castro et al (2006). However more recent work by Sun et al (2015) should be considered. 

Discussion

Pg15Para4: The authors should discuss and cite to prior studies examining wealth or income and likelihood of childhood immunization. In higher income nations it is well established that sub-populations with high levels of income are associated with lower immunization rates. An open question here (for DHS data) is whether the determination of wealth is able to distinguish such high income subpopulations in each locale that would be more likely to avoid immunization. This is a caveat on statements regarding wealth and immunization.

[LINK]

---

## [Decision Letter · Decision Letter 2]

13 Dec 2022

Dear Dr. Dimitrova,

Thank you very much for re-submitting your manuscript "Essential childhood immunization in 43 low- and middle-income countries: analysis of spatial trends and socioeconomic inequalities in vaccine coverage" (PMEDICINE-D-22-01743R2) for review by PLOS Medicine.

I have discussed the paper with my colleagues and the academic editor and it was also seen again by 4 reviewers. I am pleased to say that provided the remaining editorial and production issues are dealt with we are planning to accept the paper for publication in the journal.

[LINK]

We look forward to receiving the revised manuscript by Dec 20 2022 11:59PM.   

Sincerely,

Philippa Dodd MBBS MRCP PhD

PLOS Medicine

plosmedicine.org

Requests from Editors:

GENERAL

Thank you for your careful attention to previous editor and reviewer comments. Please address further requests below, in full.

AUTHOR SUMMARY

Line 39: “…children from pooper…” suggest “poorer”?

Please check spelling and grammar throughout for minor errors

INTRODUCTION

Suggest moving paragraph starting at lines 67-76, to the final paragraph of the introduction. The introduction should conclude with a final paragraph clearly stating the study aim.

If there has been a systematic review of the evidence related to your study (or you have conducted one), please refer to and reference that review and indicate whether it supports the need for your study.

TABLES and FIGURES

In figure 2 and 3 captions you write “inex” do you mean index? Please revise

Figure S6: it would be helpful to explicitly state in the caption that the dark dots and lines represent missed vaccines – its not easy at first glance to understand which (light Vs dark dots) represent missed vaccine doses

SOCIAL MEDIA

To help us extend the reach of your research, please provide any Twitter handle(s) that would be appropriate to tag, including your own, your coauthors’, your institution, funder, or lab. Please respond to this email with any handles you wish to be included when we tweet this paper.

Comments from Reviewers:

Reviewer #1: The authors have satisfactorily addressed all my comments on the previous version of the paper. 

I only have a minor comment that can be addressed in the limitation section of the paper. As the survey year for the countries includes the time period before and during COVID-19 pandemic (2014-2021), the vaccination rates of childhood vaccination may have impacted following the pandemic for all countries but the changes in vaccination only captured in countries with the survey years of 2020 and 2021.

Reviewer #2: I appreciate the authors' thoughtful revisions and attention to these comments. I have a few minor follow-up comments regarding the limitations, but otherwise most of my comments have been thoroughly addressed.

Major comments:

1. Previous major comment #1 (regarding uncertainty): I thank the authors for providing uncertainty estimates for the concentration indexes. As expected, some of the confidence intervals are somewhat broad, but their inclusion in the tables and figures is a substantial improvement to the paper. In particular the supplementary figure S3 - combined with the previous figures - is quite helpful. No further comments. 

2. Previous major comment #2 (regarding the need to add text describing the limitations of the analysis). The added text for the limitations is much appreciated and thoughtful. I have two follow-up questions from this added text:

2a. The authors write: "The vaccination information was verified via vaccination cards for 67% of children in the sample. For the rest, this information was based on the mother's recall, which is subject to recall bias. However, the validity of relying on parental reporting of immunization has been verified in the literature [63]"

The cited article is from a convenience sample of 108 children in a high income setting in the early 2000s; the literature on the reliability of parental recall is much more mixed than this statement would suggest. See for instance Dansereau et al 2020 (https://pubmed.ncbi.nlm.nih.gov/32270134/) which suggests that the validity of recall compared to home-based and facility-based methods varies broadly in low- and middle-income countries. I would suggest that the authors might want to capture some of this nuance in their phrasing to more completely illustrate the challenges inherent in relying on maternal recall data.

2b. The authors write: "Another limitation is that we have not been able to distinguish between vaccines administered as part of routine immunization services versus immunization campaigns, which may be less effective in the long term [64], due to lack of such data." 

Similar to my comment above, the literature on immunization campaigns is perhaps more nuanced than suggested here. For instance, Portnoy et al 2020 examined survey data, finding that supplemental immunization activities (campaigns) tended to have a more pro-equity distributional impact across wealth quintiles than routine immunization (https://www.ncbi.nlm.nih.gov/pmc/articles/PMC7519803/). The authors might consider rephrasing this section to account for this sort of nuance.

Minor comments:

1. Minor comment #1 (regarding MCV1 age of administration and the ages of analysis). I appreciate the change to the 15-35 month reference age range to deal with this limitation. As the authors note, this will improve comparability across countries although does result (likely) in more catch-up vaccination being captured, where relevant. While there is no perfect solution to these questions, though, the switch (and sensitivity analysis among 24-35 month olds in tables S3/S4) is well-reasoned and explained in the text and strengthens the paper. No further comments. 

2. Minor comment #2 (regarding RI vs campaign doses). As above, I appreciate the addition of this to the limitations section; see my comment above regarding the validity of parental recall but otherwise I have no additional comments in this regard. 

3. Minor comment #3 (regarding other types of analyses of inequality). The added text describing other dimensions of inequality in vaccination coverage is much appreciated, as is the broader appeal to available literature. I thank the authors for these additions, which increase the richness of the discussion, and have no further comments in this area.

4. Minor comment #4 (regarding language describing the scale of spatial heterogeneities assessed here). I appreciate that the authors have taken care to more accurately describe these subnational levels of analysis throughout the text, and have added the differing spatial scales across countries to the limitations of the analysis. No further comments. 

Reviewer #3: The authors have addressed my points.

Michael Dewey

Reviewer #4: The authors have largely addressed my initial concerns with their reviewer responses. What is still needed in this manuscript is for the authors to run a final check for spelling and grammar, as a few instances of both are over the top. For example, line 39 of page 3. Otherwise, given the limitations of DHS data, I believe the authors have done a credible job in their analysis and write-up.

[LINK]

---

## [Editor Report · Decision Letter 3]

28 Dec 2022

Dear Dr Dimitrova, 

On behalf of my colleagues and the Academic Editor, Professor Margaret Kruk, I am pleased to inform you that we have agreed to publish your manuscript "Essential childhood immunization in 43 low- and middle-income countries: analysis of spatial trends and socioeconomic inequalities in vaccine coverage" (PMEDICINE-D-22-01743R3) in PLOS Medicine.

Prior to publication please ensure that the final revision detailed below in made:

* Line 369-370: “…some evidence shows that such campaigns reduce health inequalities [67], while other research shows reduced effectiveness in the long term [66].” Suggest “limited effectiveness…” in place of “reduced” , or something similar given the earlier use of the same word, its appearance twice confuses the sentence a little.

PRESS

Best wishes,

Pippa 

Philippa Dodd, MBBS MRCP PhD 

PLOS Medicine